# Distribution Analysis of the Lifespan Trait in Drosophila

**DOI:** 10.3390/ijms262411987

**Published:** 2025-12-12

**Authors:** Camila A. Yumuhova, Alexander V. Konopatov, Alexander A. Shtil, Oleg V. Bylino

**Affiliations:** 1Department of Regulation of Genetic Processes, Laboratory of Molecular Organization of the Genome, Institute of Gene Biology, Russian Academy of Sciences, 34/5 Vavilov St., Moscow 119334, Russia; 2Blokhin National Medical Research Center of Oncology, 23 Kashirskoe Highway, Moscow 115522, Russia

**Keywords:** lifespan, distribution analysis, Drosophila, aging, longevity, *white* gene, survival curves

## Abstract

Research into longevity and aging involves comparing the size of cohorts at certain points on survival curves. However, this analysis is oversimplified because it provides limited information about the sample structure and the distribution of lifespan as a trait. Here, we introduce a method for estimating lifespan across the entire data range using distribution analysis. More specifically, we propose dividing the lifespan series into intervals, obtaining the frequencies of phenotypes by lifespan within the sample, followed by distribution analysis using the normality criterion. Additionally, to visualize the differences, we propose describing the resulting distributions formally using the normal distribution function and the β-distribution function. We demonstrate that the proposed methodology enables to extract additional information from survival data, providing new insights into the processes that occur in populations in response to genetic interventions and shedding light on their impact on ontogenesis. In particular, we observed that the lifespan distribution in Drosophila may not meet the normality criterion and may take different shapes depending on the line’s genotype or in response to genetic interventions. The proposed approach adds a new layer of information to studies of longevity and aging and expands the toolkit of methods used to analyze survival data.

## 1. Introduction

‘I think it may fairly be assumed, in the light of what we now know, that no other measure will, statistically speaking, furnish so delicate and precise a measure of the general constitutional fitness of individuals as will their duration of life.’Raymond Pearl, 1923

Lifespan is one of the most complex quantitative traits. Like all quantitative traits, lifespan has a distribution [1,2]. It is fundamental to understand how the distribution of lifespan in study populations will change in response to various interventions designed to increase lifespan (genetic, pharmacological, environmental), and to be able to correctly assess, understand, and interpret these changes. Studying of regularities of changes in lifespan distribution is a prerequisite for extending human lifespan as a species [3]. This kind of research is conveniently carried out using short-lived model organisms such as the fruit fly Drosophila or the worm *Caenorhabditis elegans*.

Drosophila is widely represented in longevity research and is used to study the effects on lifespan of various interventions, including environmental factors (e.g., housing/lifestyle, diet, temperature, oxidative stress, etc.), pharmacological factors, and genetic factors (e.g., introduced mutations or transgenes) [2,4,5,6,7,8,9,10,11]. Surprisingly, the survival patterns in Drosophila mirror those in humans. This phenomenon was remarkably discovered at the beginning of the 20th century by the distinguished researcher Professor Raymond Pearl [12], who proposed using lifespan as the main indicator of fitness [13]. Although the number of offspring produced is currently considered the main criterion for genotype fitness [14,15,16], lifespan is nevertheless one of the main components of an individual’s Darwinian fitness (life cycle traits), along with age of sexual maturity, fecundity, fertility and age-related survival dynamics [17,18]. Thus, lifespan is, if not the most important, then at least one of the most important components of overall fitness.

Studies of longevity and aging are inherently linked to the generation of Kaplan–Meier survival curves [19,20,21]. A survival curve reflects the decline of a cohort (individuals born at the same time) or population over time. Survival curves are commonly used in clinical studies, for example, to monitor the survival of patients after chemotherapy or cardiological events, as well as in longevity studies to track the mortality of short-lived organisms in response to a variety of interventions. To assess the differences between survival curves, a comparison of two samples based on Median (50th percentile of mortality) and maximum (90% of mortality) lifespans is usually used. The former is commonly evaluated using relevant statistical tests, such as the Kolmogorov–Smirnov two-sample test [22] or the log-rank test (also known as the Mantel–Cox test) [23,24], and the Gehan–Breslow–Wilcoxon test [25,26,27,28]. To compare the latter, the Wang–Allison [29] (the main equation in this paper was subsequently corrected, and the correct equation was published earlier by Mehrotra et al. on page 443 [30]) and the chi-squared test [31,32] are more commonly cited in scientific literature.

In addition to survival curves, predictive analysis methods are used, such as the Cox proportional hazards regression model [33,34,35,36,37], which deals with the hazard function, or the Accelerated failure time model (AFT model) [38,39], which deals with survival time/curve and some others (for more information see [38]). The Cox model is fundamentally based on the Gompertz equation, or the Gompertz–Makeham equation [40], in which medical risks, for example, are substituted for the risk factor, x (time). The AFT model is a more sophisticated and flexible modification of the Gompertz principle. The Cox model is prevalent in medical statistics and is also widely used in scientific literature [41,42,43,44]. The AFT model is used in scientific research on longevity [45] and often yields exceptional results when employing Weibull’s distribution [46].

However, these tests only provide information about the significance of differences or risks at specific points in time. Thus, neither survival curves nor mortality curves offer detailed insights into the frequency of phenotypes by lifespan within the sample. Instead, they only provide a general idea of how mortality or risks increase over time in the population under study. This requires the development of new approaches to the presentation and analysis of survival data that clearly show differences in the structure of the samples being compared.

Here, to solve the problem of sample structure, we converted the daily mortality data (deaths per day) into a lifespan series, where the unit of measurement is the lifespan of each individual. Then, to smooth out daily mortality fluctuations, we combined mortality data from the previous and following days. We divided the survival data into intervals/phenotype frequencies according to Sturges’ rule and chose a convenient interval size of five days. Overlaying the probability series obtained in this way (i.e., phenotype frequencies by lifespan) for different samples and formally describing them using normal distribution curves made it possible to clearly compare the samples.

We analyzed the normality of the initial and obtained interval distributions using two tests: the Kolmogorov–Smirnov test for single samples (KS test) and the Shapiro–Wilk test (SW test). Of all the tested options, the SW test on interval data proved to be the most effective, sensitive, and informative. The optimal conditions for applying the SW test to survival data were a sample size of up to 200 individuals and dividing the lifespan data of the compared samples into an equal number of intervals, calculated using Sturges’ rule. This allows for fractional and unequal interval sizes, but requires an equal number of intervals between samples. To visualize the frequency distributions of the samples, we used the β-function in addition to the normal distribution function. Using the β-function, we demonstrated that distributions in Drosophila can take various forms, including a left-skewed β-distribution, a right-skewed β-distribution, a symmetric bell-shaped distribution similar to a normal distribution, and a uniform (plateau-like)/linear distribution. Using quantitative criteria of normality allows one to detect subtle changes in the sample structure, even when the shape and/or position of the survival curves of the compared cohorts are similar. It has been demonstrated that genetic interventions (introduction of the white mutation) are accompanied by changes in phenotype frequencies within the sample, changes in distribution type, and changes in quantitative parameters of normality. Additionally, it has been established that the distribution type in control WT-like lines does not necessarily correspond to the normal distribution. The distribution of these lines may be completely different. Thus, the presented study covers the entire spectrum of questions regarding the analysis of the distribution of lifespan in Drosophila.

## 2. Results

### 2.1. Dividing Survival Data into Intervals Enables an Effective Assessment of Phenotype Frequencies by Lifespan Within the Sample

To see the distribution of the lifespan trait in the studied samples, we converted the daily mortality series (Figure 1a) into a lifespan series where each value represents an individual’s lifespan (Figure 1b).

To understand the nature of the distribution, we applied normality tests to the resulting distributions using the standard KS and SW tests for analyzing single samples. Unexpectedly, the results of the analysis showed that the SW test indicated a non-normal distribution for all 24 samples, while the KS test showed that the lifespan data for six of the 24 samples were normally distributed (Table 1, Table 2 and Table 3, columns 6 and 10).

The results obtained led us to analyze the reasons for the unsatisfactory outcome. A thorough review of the experimental data showed that the survival data often contain mortality dips (days with zero mortality) and mortality outliers (days with a sharp increase in mortality). A typical example is shown in Figure 1a. We hypothesized that fluctuations in daily mortality might affect the assessment of normality.

To mitigate the effect of mortality fluctuations, we combined mortality of adjacent days by dividing the lifespan series into intervals according to the method of Sturges (the procedure is described in the Section 3.2 of the Materials and Methods), who proposed a rule of thumb for determining the size (class) of an interval [47]. We applied Sturges’ formula to our data with further rounding to integers (according to rounding rules) and obtained the following interval sizes for different samples: 7,7,7,7,6,5,6,5 (w^1118^ 2024), 8,8,9,8,8,8,7,7 (w^67c23^ 2012), 7,6,8,8,7,6,8,7 (w^67c23^ 2024) (the order of the samples is the same as in Table 1, Table 2 and Table 3). For a correct comparison, the same interval size must be used. According to Sturges, the most “convenient” intervals are 2,5,10,20,50,100,200, etc. When the interval calculated by Sturges’s rule does not match a convenient interval, the nearest convenient interval should be used. We settled on an interval size of 5 days (Figure 1c and Appendix A), testing a range of 3–7 days. We can see how the distributions with 79 and 73 days of observation (Figure 1a and Appendix A) transform into distributions with 15 and 14 intervals (Figure 1c). The interval size of 5 days ensured acceptable detail and similarity to the original death-per-day distributions (compare Figure 1a with Figure 1c and Appendix A). The seemingly convenient value of 7 days created an overly simplistic, rarefied picture of the graph (compare Figure 1a and Appendix A with Appendix A). Thus, by dividing the lifespan series into equal intervals, we obtained a series of probabilities, i.e., phenotype frequencies by lifespan.

In graphs, mortality can be expressed in absolute numbers (i.e., the number of individuals) or in relative units (i.e., percentages or probabilities). Comparing in relative units normalizes the size of the samples being compared and appears to be the most informative. If one of the samples has a shorter lifespan or if there is no mortality during the initial time interval (five days), additional intervals with zero probability of mortality can be added to the obtained probability series. This makes it possible to visually compare the two samples in graphs. However, it is important to note that all zero intervals, both at the beginning and end of the distribution, should be excluded from further KS and SW test calculations, as including them introduces distortions to the values of the test statistics and affects the *p*-value.

To facilitate visual comparison of the samples, we superimposed the original distributions of deaths per day (Figure 1a and Appendix A) or the obtained distributions of lifespan probabilities (Figure 1c,d and Appendix A) with the corresponding normal distribution curves (indicated by the bell-shaped curves in Figure 1a and some subsequent figures). These curves were calculated based on the average lifespan and standard deviation of the original (non-intervalized) lifespan series. The procedure is described in the Section 3.3 of the Materials and Methods. This allowed us to compare two simple-looking normal distributions rather than two complex-looking frequency series and formalize and quickly understand the difference between the two samples.

Thus, dividing lifespan data into intervals and superimposing normal distribution curves on the resulting probability series eliminates the problem of mortality fluctuation, clearly shows differences in phenotype frequencies by lifespan within a sample, and makes it possible to effectively compare samples with each other.

### 2.2. Using Kolmogorov–Smirnov and Shapiro–Wilk Normality Tests to Describe the Parameters of the Lifespan Distribution

Next, we analyzed the resulting intervalized distributions using the KS and SW test and compared the results with the results of the analysis of the original distributions (not divided into intervals) obtained using the same tests for all 24 samples (Table 1, Table 2 and Table 3). We determined test statistics (numerical quantitative description of the results) and normality (qualitative description of the results according to the “yes” and “no” principle). Similarly to the SW test on non-intervalized data, the KS test on intervalized data was found to be unrepresentative and showed a pattern of normality for all 24 samples (Table 1, Table 2 and Table 3, column 12). Applying the Lilliefors correction for small samples for the KS test helped to reveal non-normality of the distribution for 8 samples (Table 1, Table 2 and Table 3, column 12). The intervalized data analyzed by the SW test differed in 4 of the 24 samples from the results on the intervalized data obtained using the Lilliefors-corrected KS test. Moreover, for 3 of these 4 samples, when the SW test identified the distribution as non-normal, the Lilliefors-corrected KS test considered it normal and only for 1 sample the opposite situation was observed (Table 1, Table 2 and Table 3, compare columns 8 and 12, cases of differences are marked in italics). Taking into account the fact that KS test without Lilliefors correction always shows normal distribution for intervalized data, and with Lilliefors correction in 3 out of 4 samples shows the second kind of error (does not find deviations from normality where they are, compared to SW test), we concluded that SW test is more accurate and representative when analyzing lifespan data divided into intervals.

Next, to clarify which type of data is preferred for analysis, we compared the results of the SW test on intervalized data with the results of the KS test on non-intervalized data. We found that there are 8 out of 24 samples where differences in the results of the two tests are observed (Table 1, Table 2 and Table 3, compare columns 8 and 10, cases of differences are in bold). In all these cases, when the SW test on intervalized data showed a normal distribution, the KS test on non-intervalized data showed a non-normal distribution. To understand why this is observed, we took a closer look at the performance of the KS test in these 8 samples.

The analysis showed that, in general, the estimation of the normality of the distribution using the KS test is strongly influenced by the variation in daily mortality. We observed that the point corresponding to the supremum of the KS function (based on the value of which the test concludes whether the distribution is normal or non-normal) can correspond to (i) a day with high mortality preceded by one or more days with low mortality (Appendix A). In this case, there is a strong rightward shift of the function, and the difference between the experimental and control CDFs is negative, (ii) a day with high mortality preceded by several days with increasing mortality (Appendix A). In this case, there is a strong rightward shift in the function, and the difference between the experimental and control CDFs is negative, (ii) a day with high mortality preceded by several days with increasing mortality (Appendix A). In this case, there is a strong leftward shift in the function, and the difference between the experimental and control CDFs is positive. Thus, in all 8 cases considered, the KS test identifies the distribution as non-normal based on the violation of the smoothness of the function.

To gain a deeper understanding of the KS test’s regularities, we performed additional experiments (Section 2.2 “Additional observations regarding the mechanics of the KS test” in Appendix A) and found that the KS test is strongly affected by fluctuations in mortality that fall in the middle of the distribution rather than at its edges, as well as by sample size *N* (Appendix A).

Thus, the results presented here and in Appendix A allow us to conclude that it is better not to use the KS test to estimate the pattern of lifespan distributions, regardless of the type of data (intervalized or non-intervalized). The KS test has low sensitivity for intervalized data and too high sensitivity for non-intervalized data. Thus, to estimate the pattern of lifespan distributions and predict biological regularities, we further used the SW test on intervalized samples.

### 2.3. Peculiarities of SW Test Functioning Under Different Intervalization Conditions and on Samples of Various Types

Before conducting experiments using the SW test to compare mutant and control line samples, we examined the methodology more closely. We tested the limitations of the SW test on two sets of data: (1) 24 real samples and (2) an ideal normal series constructed using experimental sample data (Yumuhova et al., in preparation; Appendix A). The ideal normal series uses information about the initial and final mortality dates, as well as the sample size (*N*). Third, we examined the SW test in simulations. In these simulations, we used ideal normal series with lengths ranging from 50 to 84 days (the range of maximum lifespans in our experiments) and sample sizes ranging from 100 to 795 individuals. In all three cases, we shifted the start of intervalization (and the initial interval) by one, two, three, or four days. In other words, we compared the results of the test on samples with initial intervals starting from zero to four days, one to five days, two to six days, three to seven days, and so on until the end of the sample. In every assay, we used an interval length of five days.

It turned out that for experimental samples that were initially normal (for which we determined a normal distribution when we performed intervalization from day zero, Table 1, Table 2 and Table 3), shifting the first initial day of intervalization from 0 to 1, 2, 3, or 4 days led to a change in the distribution type to non-normal in 12.5% of cases (Table 4). For non-normal samples, shifting the initial day of intervalization changed the distribution type in 10.7% of cases (Table 4). For an ideal normal series constructed based on experimental sample data, shifting the initial day of intervalization from 0 to 1, 2, 3, or 4 days resulted in 75.8% of the 120 simulated samples (24 × 5 days) being normal and 24.2% being non-normal (Appendix A). In these experiments, the number of intervals depending on maximum lifespan (MaxLS) varied from 10 to 17. Thus, even with ideally normally distributed data, a false negative result may occur in a certain percentage of cases when the initial day of the initialization is shifted (the normal distribution must be re-determined, but a non-normal distribution is determined instead).

During the simulations, we examined how sample size (N) and lifespan duration affected the SW test results. When studying the influence of N, we found the following: As *N* increased from 100 to 795 (the maximum recorded in our experiments), the proportion of normally distributed samples decreased from ~95% at *N* = 100 to 34–74% at *N* = 795, depending on when intervalization started (days 0, 1, 2, 3, or 4) (Figure 2a). The average *p*-value decreased as *N* increased, indicating a trend toward greater non-normality (Figure 2b). Thus, as *N* increased, the number of false negative results increased. At the same time, the number of false negative results did not increase; it remained stable and was in a range of 1–6%, depending on the day on which the intervalization was started with *N* of 100 to 200 individuals (Figure 2a). Therefore, to minimize SW test error on lifespan samples, the sample size should be between 100 and 200 individuals.

When studying the effect of lifespan duration on the SW test result, the following was found: As the lifespan increased from 50 to 84 days (the range of MaxLS in our experiments), the proportion of normally distributed samples decreased. Depending on the day when intervalization was started, the proportion ranged from 19.8% (day 0) to 69.3% (day 4) (Figure 2c). The graphs in Figure 2c show a pronounced periodic alternation of sharp decreases and smooth increases in the number of normally distributed samples, with a fluctuation period of five days. This graph pattern is likely due to the formation of an additional interval when the lifespan exceeds the limit of the last interval. This results in a sharp decrease in the proportion of normally distributed samples as the number of intervals increases by one. The opposite trend, a gradual increase in the proportion of normally distributed samples after the sharp drop, can be explained by the gradual increase in size (height) of the last interval as lifespan increases. The average *p*-value decreased as lifespan increased, tending toward greater non-normality (Figure 2d). Thus, an increase in lifespan leads to the SW test more often “perceiving” the distribution as non-normal.

Thus, an increase in lifespan with a constant interval length of five days will be accompanied by a corresponding increase in the number of intervals. This leads to an increase in the proportion of samples with non-normal distributions. Consequently, as *N* increases (the number of intervals/frequencies of phenotypes in terms of lifespan in this case), the probability of determining the distribution as non-normal using the SW test increases. This assumption is also supported by the fact that changing the first day of intervalization (0, 1, 2, 3, 4)—which is obviously accompanied by a reduction in the total number of intervals—also results in a decrease in the number of non-normally distributed samples (Figure 2c).

To investigate this issue, we conducted simulations and compared the results of the SW test when we intervalized an ideal normal series of varying lengths into 5-day intervals or when we intervalized this series into the optimal number of intervals according to Sturges’ rule (see the Section 3.2 of Materials and Methods). For each simulation, we took the result for the entire set of samples, as described in the legend for Figure 2. The only difference was that the *N* ranged from 100 to 940. We found that, when intervalizing by five days, the number of normally distributed samples was 63.29%. Whereas when intervalizing by Sturges, it is, depending on the type of rounding, 98.92% (rounding up, accompanied by a decrease in the number of intervals), 93.05% (rounding according to the rounding rule, giving an intermediate value), or 86.22% (rounding down, accompanied by an increase in the number of intervals). Thus, (i) partitioning an ideal normal lifespan series into intervals according to Sturges’ rule gives a higher percentage of preservation/retention of normal distributions than partitioning it into 5-day intervals, and (ii) the result of the SW test is sensitive to the number of intervals.

In summary, the results of studying the SW test’s functioning on small samples allow us to draw several conclusions: (1) The SW test result is influenced by lifespan duration (affected by the number of sample intervals) and sample size. The greater the number of intervals or *N*, the greater the number of false negative results. (2) Error in predicting normality caused by *N* can be minimized by using a sample size of up to 200 individuals. (3) Error arising from an increased number of intervals (e.g., intervalizing every five days to obtain a frequency series of lifespan) can be minimized by intervalizing according to Sturges’ rule. The number of intervals will be smaller, and the interval size will be larger. At the same time, it is clear that the number of intervals in the compared samples must be the same.

### 2.4. The Normality Criteria, the Beta Function, and the Normal Distribution Function Can Be Used to Describe the Effect of Genetic Interventions on the Distribution of Lifespan

We analyzed survival data for control lines and lines undergoing genetic interventions to investigate whether the normality criterion can be used to identify biological patterns and find differences between samples. The white mutation was used as a model. White mutants had the same genetic background as WT-like control flies. We used both quantitative (W, *p*) and qualitative (non-normal/normal) components of normality.

The effect of two different classical alleles of the gene, w^1118^ and w^67c23^, on lifespan was investigated. Estimating the effect of the *white* gene on lifespan is an important task because, despite more than 115 years of studying this gene [48,49], the physiological significance of *white* in Drosophila remains unclear [50,51,52,53,54].

The data obtained in the previous section clearly show that the number of intervals affects the normality assessment result for small samples using the SW test. Therefore, to use the SW test as a tool for identifying biological patterns and comparing two samples based on normality criteria, we should: (1) equalize the number of intervals between samples and (2) reduce the number of intervals. Accordingly, we abandoned the division of survival data into 5-day intervals and instead re-intervalized the samples according to Sturges’ rule, using a fractional interval size (Table 5). Next, we compared what would happen to the experimental samples in a paired comparison of mutant-control when we intervalized the data for five days and according to Sturges’ rule. When equalizing the number of intervals, we used the number of intervals of the sample with the smallest value in each pair. As shown in Table 5, the same number of intervals for the mutant and control, coupled with a decrease in the total number of intervals, causes some non-normally distributed samples to become normally distributed when partitioned by Sturges’ rule. However, the opposite pattern is sometimes observed.

In an experiment with the w^1118^ 2024 allele (which reduces lifespan), it was found that the distribution changed significantly in three out of four cases when the *white* mutation was introduced (Figure 3a,b,d). The survival curves of the mutant and control differed significantly in all these cases according to the two-sample KS test (Table 1, column 13). A minor change in distribution shape also occurred in the Oregon RC male vs. white Oregon RC male pair (Figure 3c). In this case, however, the two-sample KS test found no differences between the survival curves. There were also almost no differences in SW test data between the Oregon RC male and the white Oregon RC male (Figure 3c). The SW test showed changes in W and *p* in three cases (Table 5, columns 4 and 5; Figure 3a,b,d).

In the Canton S genotype, there was a change in the distribution type (normal/non-normal) between the mutant and the control (Table 5, columns 4, 5). Thus, changes in survival curves correlate well with changes in distribution parameters according to SW test data, and in cases where survival curves do not differ, the SW test also shows no differences.

In the experiment with the w^67c23^ allele (which increases lifespan), the survival curves of the mutant and control groups differed significantly from each other according to the two-sample KS test in all cases (Table 2, column 13; Figure 4). The distribution changed in all cases and depended strongly on the genotype of the line. For instance, on the Canton S genetic background, the frequency distribution in both males and females was described by an arcuate β-distribution, which had a pronounced Mode (Figure 4a,b). When the mutation was introduced, the arcuate β distribution transformed into a uniform β-distribution, and the frequency distribution lost its pronounced Mode (Figure 4a,b). On the Oregon RC genetic background, the shape of the frequency distribution did not change significantly when the mutation was introduced, retaining a pronounced Mode (Figure 4c,d). Changes in the shape of the frequency distribution were accompanied by changes in W and *p* in all cases (Table 5, columns 4, 5). The distribution type changed to the opposite for the Canton S genotype but not the Oregon RC genotype (Table 5, columns 4, 5). Thus, an increase in lifespan in response to genetic intervention is accompanied by genotype-dependent changes in the nature of frequency distribution, β-distribution curves, and quantitative measures of normality.

In an experiment with the w^67c23^ allele (2012), it was found that for the same lines in the collection and using the same algorithm for introducing the w^67c23^ allele into the lines, a different distribution pattern was observed than 12 years later (2024) (compare Figure 4 and Figure 5). The 2024 and 2012 experiments were performed at slightly different temperatures: 25 °C and 23 °C, respectively. Therefore, it is not possible to directly compare the two experiments. However, it is possible to judge the changes that have occurred in the lines based on the ratio of effects between the control and the mutant. According to the two-sample KS test, the positive effect of the w^67c23^ allele on lifespan was observed in the 2012 experiment for only two of the four genotype pairs (Figure 5a,d). In contrast, in 2024, the positive effect was evident for all genotypes studied (Table 3, column 13, and Table 2, column 13). In 2012, the frequency distributions of the mutant and control did not differ fundamentally in shape, and the mutation only modified the frequency distribution without changing its shape. In contrast, the 2024 experiment revealed a sharp change in the frequency distribution’s shape (at least for the Canton S genotype). The distribution type according to SW test data changed in 2012 for two cases (Figure 5a,d), and in 2024 also for two cases (Figure 4a,b). However, the direction of change for the Canton S males (Figure 4a) was in the opposite direction. The quantitative parameters of normality, W and *p*, changed when the mutation was introduced in both 2012 and 2024 (Table 5, columns 4 and 5). Based on the relationship between the survival curves and the frequency distributions of the mutant and control lines in 2012 and 2024, it can be assumed that the genetic background of the laboratory lines has changed so much that it has modified the effects of the w^67c23^ allele on lifespan distribution.

In all three experiments, the distribution of phenotype frequencies by lifespan can be formally described using either the β-function or the normal distribution function (Appendix A). While this representation is more unified, as two bell-shaped curves of the same type overlap, it is less effective at showing the differences between samples than the β-distribution.

In summary, the results of the analysis of three experiments allow a number of conclusions to be drawn:

(i) Determining quantitative indicators of normality (W and *p*) using the SW test (calculating the interval size according to Sturges’ rule and equalizing the compared samples by the number of intervals) enables subtle changes in the sample structure to be detected, even when the survival curves of the compared cohorts are almost identical in shape and do not differ significantly according to the results of statistical curve comparison tests (e.g., the two-sample Kolmogorov–Smirnov test; experiment with allele w^1118^ 2024).

(ii) The characterization of the distribution structure (frequency/probability series of phenotypes by lifespan) combined with a formal description of the distribution using the β-function or the normal distribution function, provides a clear picture of the processes occurring in the studied populations in response to genetic interventions. The β-distribution more accurately represents the shape of the frequency series. However, the normal distribution is more consistent because it has the same shape for both samples being compared.

(iii) The distributions of the lifespan of laboratory (highly inbred) Drosophila lines differ in form, depending on the genotype of the line and the year of the experiment, and change with the introduction of mutations. As can be assumed a priori, the distribution type in control Drosophila lines does not necessarily have to meet the normality criterion; the distribution of wild-type lines may be non-normal (experiments with alleles w^1118^ 2024 and w^67c23^ 2024).

## 3. Discussion

### 3.1. The Situation with the Study of Lifespan Distributions in General

It is believed that human lifespan in heterogeneous populations is normally distributed in large data sets, but is skewed to the right [55]. At the same time, it is obvious that the rightward skew of human lifespan distributions (i.e., toward later ages of mortality) is a consequence of the overall level of development of human culture, living conditions, and medicine, and this situation has nothing to do with the survival curves of humans and apes in the wild [56]. In this regard, any “natural” patterns in changes in human lifespan cannot be understood easily by studying mortality statistics in human populations or even in populations of other mammals [57,58,59]. In this regard, any “natural” patterns in changes in human lifespan cannot be understood easily by studying mortality statistics in human populations or even in populations of other mammals. Therefore, it is clear that observational experiments are not sufficient and that intervention experiments on shorter-lived subjects are absolutely necessary.

The distribution of lifespan in *C. elegans* is also considered normal. However, as with humans, it can be skewed depending on the conditions in which the worms are cultivated [46]. Therefore, *C. elegans*, being the shortest-lived multicellular model organism commonly used in laboratories, are a promising subject for studying lifespan distributions. However, aging in *C. elegans* is highly plastic and can be easily delayed, as these worms possess a programme of age-related suicide [60,61]. Perhaps aging in worms is even more complex, as it can be represented by two components that can be isolated using mathematical approaches and various cultivation conditions [46]. For these reasons, studying patterns of change in lifespan distribution in worms requires caution. On the other hand, the lifespan distribution in worms changes in response to the introduction of mutations, in a manner similar to what we observed in Drosophila under our experimental conditions [62]. For example, when the right tail of the distribution shifts to the right as the lifespan increases, the clearly defined mode disappears, and the distribution becomes plateau-like (uniform) (Figure 4a,b).

Although scientific literature contains many descriptions of experiments on the lifespan of Drosophila [63], accompanied by survival curves, there are virtually no descriptions of lifespan distribution in Drosophila. Even fewer studies have examined the patterns of change in these distributions. The literature contains only fragmentary references stating that the lifespan distributions in Drosophila are normal [64]. In this study, we demonstrate that this is not the case. Our data on the distribution of phenotype frequencies by lifespan show that the lifespan trait can be distributed very diversely in the studied samples, depending on the genetic background of the line. There are normal and non-normal distributions; distributions with high initial mortality and high late survival; and distributions resembling uniform (plateau-like) or linear ones without a pronounced bell-shaped rise in the center (Figure 5d, for example). The frequencies of phenotypes by lifespan are distributed uniquely and characteristically for each line.

### 3.2. The Destabilisation of Ontogenesis and Changes in Lifespan Distribution Under Genetic Interventions

Having analyzed the distribution of Drosophila lifespan series, we concluded that introducing mutations leads to changes in the frequencies of short- and long-lived phenotypes across the entire frequency range, relative to the control sample (Figure 2, Figure 3 and Figure 4). This was observed when introducing mutations that increased (allele w^67c23^) or decreased (allele w^1118^) lifespan. Thus, a general destabilization of ontogenesis is manifested, which is expressed in a change in the frequencies of phenotypes across the entire distribution.

Furthermore, our data show that genetic interventions (in this case, the *white* mutation) interact differently with the genotypes of different laboratory lines. Therefore, the mechanism by which the mutation affects lifespan may vary depending on the genetic background. Due to genetic drift in small, isolated populations, laboratory Drosophila lines accumulate genetic changes independently. This means that the interaction between a known mutation and a set of mutations unique to each line affects ontogenesis in its own way.

In this study, we did not specifically examine the asymmetry or kurtosis of distributions. This is a distinct task that requires its own research and in-depth analysis. The mathematical apparatus involved is quite complex [65]. However, a brief examination of the graphs obtained using our methods reveals the following:

(i) The introduction of the w^1118^ 2024 allele (reduces lifespan) shifts the distribution to the left relative to the control (Appendix A), whereas the introduction of the w^67c23^ 2024 allele (increases lifespan) shifts the distribution to the right (Appendix A).

(ii) When lifespan decreases, the distribution as a whole becomes narrower and higher (Appendix A, σ decreases), or shifts without changing fundamentally (Appendix A). Conversely, when lifespan increases, the distribution shifts but does not fundamentally change (Appendix A), or it becomes wider and flatter (Appendix A, σ increased). All of this reflects changes in the distribution in response to genetic interventions.

### 3.3. Selection of Tests for Analysing Lifespan Data to Determine Normality

At least 40 statistical tests have been developed to analyze the distribution of single samples [66]. The main criteria for the tests that are used are convenience (ease of calculation) and indicativeness (the ability to identify patterns). The most popular and widely used tests are the Kolmogorov–Smirnov test (KS test) [67,68] and the Shapiro–Wilk test (SW test) [69,70]. Both tests satisfy both criteria in their classical versions.

***The KS test*** is a universal test that can be used to check whether a set of data belongs to a particular distribution, such as uniform, non-uniform, normal or non-normal, or exponential or linear. To assess normality or non-normality, the cumulative density function (CDF) should be used. Other functions are used for other types of distribution (e.g., the linear function). This test can be used with both discrete and continuous data.

The *p*-value of the KS test is typically calculated using the formula originally developed by Kolmogorov in 1933 [71,72,73]. As an alternative, a number of more contemporary and efficacious formulae (simpler to calculate) have been proposed in the literature. For instance, the formula advanced by Marsaglia [74]:(1)$$P=2*e^{-\left(2.000071+\frac{0.331}{\sqrt{N}}+\frac{1.409}{N}\right)*N*\left(D^{2}\right)}$$
where *P* is the *p*-value, *e* is the base of the natural logarithm, *N* is the sample size, and *D* is the KS test statistic.

We tested this formula and compared its performance with Kolmogorov’s classic formula for intervalized and non-intervalized lifespan series for our experimental samples (Appendix A). It turned out that in the case of non-intervalized series (samples with large *N*), the discrepancy in *p*-values between the two formulas is observed in the third decimal place, i.e., both formulas are equivalent. However, for intervalized series (small samples), Marsaglia’s formula sometimes gives *p*-values greater than 1 (e.g., 1.391) (Appendix A). Therefore, Marsaglia’s formula is not applicable when *N* < 100; instead, Kolmogorov’s formula should be used. For *N* < 50, Lilliefors’ percentage point table is required [75].

***The SW test*** is based on Johnson’s approximation method using the SB model, which approximates the observed distribution to a normal distribution [76]. In their 1965 paper, Shapiro and Wilk calculated critical *p* values for specific percentage points (0.01, 0.02, 0.05, 0.10, 0.50, 0.90, 0.95, 0.98 and 0.99) for *N* ranging from 3 to 50 [77]. This is a significant drawback of the original version of the SW test (generalized SW test) for lifespan studies since these studies involve analyzing much larger data sets. Therefore, for samples of size greater than 50, the Royston method can be used. This method proposes a modified formula for calculating the SW test statistic W, new coefficients for calculating W and an algorithm for calculating the *p*-value for any sample size [69].

Like Johnson, Royston transforms the observed distribution z to normal, but uses a more sophisticated method of calculating W than in the generalized SW test. This method employs a function containing a polynomial to calculate W and *p*. The procedure for calculating W using Royston’s method is the same for all sample sizes from 12 to 5000. However, it differs for samples smaller than 12, for which other formulas are used to find g(X), µ_z_ and σ_z_. However, despite Royston’s method being widely used in online calculators, upon closer inspection, the Royston calculation procedure was found to be too cumbersome and does not meet the ease-of-use criterion. Therefore, the complexity of the calculation system and the associated risk of technical errors preclude the use of this method in thorough manual analysis of lifespan data series. Furthermore, when we calculated W and *p* for non-intervalized lifespan series using the Royston method, we found that all the analyzed distributions were non-normal (Table 1, Table 2 and Table 3, columns 5 and 6). Therefore, for original lifespan data not divided into intervals (i.e., samples with large *N*), the SW test in Royston’s modification is inconclusive. Therefore, in order to analyze lifespan series, we proposed dividing survival data into intervals and using the W calculation system from the standard generalized SW test. We combined this with the Royston method’s procedure for accurately calculating *p* for any sample size. This approach overcomes the limitation of a sample size of 50 values and allows the use of a simple W calculation system based on the original work by Shapiro and Wilk.

***The division of survival data into intervals*** and the pooling of mortality on adjacent days, which we propose as a means of overcoming the problem of daily mortality fluctuations, essentially corresponds to the counting of dead individuals every 2–3 days of life [4,78,79] or even once a week [80], which is acceptable in lifespan studies. In addition, combining mortality on adjacent days eliminates several shortcomings of daily counting:

(i) inaccuracy in determining the date of death. For example, an individual may die immediately after testing on the same day but be recorded as having died the next day.

(ii) the stochastic effect of transferring flies to new food. For example, if the food spoils prematurely, it is replaced earlier than after three days, which may affect lifespan. Clearly, combining the mortality of adjacent days corrects for these effects.

According to our data from 24 samples, the maximum lifespan range (i.e., the age at which 100% of individuals die) of laboratory Drosophila lines is 50–84 days at 25 °C. Therefore, when divided into 5-day intervals to determine frequencies, there will be 12–17 analyzed segments/intervals/frequencies of phenotypes by lifespan. If one of the samples differs in the number of intervals, zero intervals (columns on graphs with a value of 0) may be introduced for the convenience of sample comparison on graphs. However, when calculating KS and SW tests, zero intervals (initial intervals that do not contain dead individuals) must be strictly excluded. For example, in the case of an additional zero interval(s) for the KS test, the value of D_n_ will change since the CDF values will shift relative to the Rank/*N* function, and D*_crit_* will decrease because, as *N* increases, the value of √*N* used to find the *p*-value will also increase (Appendix A). For the SW test, the values of the W test statistic and *p*-value will change since an additional zero interval leads to a change in the number and composition of the coefficients used to calculate W and *p* (Appendix A).

When conducting simulations, we found that even when using perfectly normal data, frequency extraction (i.e., the intervalization of the lifespan series) leads to the SW test producing some level of error when predicting normality (see the Section 2.3). Consequently, the SW test is not an absolutely ideal tool for accurately predicting normality in small samples. However, according to our data, the SW test is more accurate with intervalization than without, and more accurate than the KS test with or without intervalization (see the Section 2.3). We do not rule out the possibility that other tests for assessing normality for small samples may exist that are more efficient. This issue requires further research. Perhaps new tests based on principles of normality assessment other than the classic SW and KS tests need to be developed for small samples.

## 4. Materials and Methods

### 4.1. Drosophila Lines and Survival Data for Analysis

For analysis, we used the *Drosophila melanogaster* lifespan series: 5, 6, 6, 6, 6, 6, 6, 6, 6, 6, 6, 6, 6, 8, 9, 9, 9, 9, 10, etc., where each number represents the lifespan (in days) of one individual fly. One hundred percent mortality was observed between days 50 and 84. We investigated WT-like laboratory control lines, Canton S and Oregon RC, as well as experimental Canton S and Oregon RC lines containing mutant alleles of the *white* gene (w^1118^ and w^67c23^). The w^67c23^ allele was isolated from the y^1^w^67c23^ standard laboratory line chromosome by recombination with the control lines’ chromosomes. The w^1118^ allele was taken from line #3605 (Bloomington Collection). The genetic backgrounds of the mutant lines were equalized through 10 backcrosses with the control lines [4,5,6,8]. The backgrounds of control lines originated from the Drosophila collection at the Institute of Molecular Genetics of the Russian Academy of Sciences.

The analysis used lifespan data from three experiments: (1) Spring 2024: Canton S, w^1118^ Canton S, Oregon RC, w^1118^ Oregon RC lines. (2) Fall 2024: Canton S, w^67c23^ Canton S, Oregon RC, w^67c23^ Oregon RC. (3) Fall 2012: Canton S, w^67c23^ Canton S and Spring 2013: Oregon RC, w^67c23^ Oregon RC. The 2024 experiments were conducted at a temperature of 25 °C in a “warm” room, and the 2012–2013 experiments were conducted at a temperature of 23 °C “on the table.” The experiments with the w^67C23^ allele, conducted in 2012–2013 (hereafter referred to as 2012) and 2024, can be considered independent experiments on different genetic backgrounds. The genetic backgrounds of the lines changed significantly during the 12-year period they were kept in the collection, as seen in the shape and pattern of the survival curves. All experiments were conducted under natural light and humidity conditions. In each experiment, males and females were kept separately. Flies were mobilized for fresh food once every three days. The experiments were carried out in 100 × 25 mm polypropylene vials using the following medium: 66 g/L of pressed yeast, 35 g/L of semolina, 60 g/L sugar, 0.5% agar-agar, 0.5% propionic acid, and 30 g/L of raisins that had been twisted through a meat grinder.

### 4.2. Partitioning Survival Data into Intervals (Obtaining Frequency/Probability Series of Lifespan)

Sturges’ rule [47] was used to determine the size of each interval:(2)$$C=\frac{R}{1+3.322\times \mathrm{lg}N}=\frac{R}{1+\log_{2}N}$$
where *C* is the optimal interval size (in our case, the number of contiguous days combined), R is the difference between the maximum and minimum sample values (in our case, maximum and minimum lifespans), *N* is the sample size (number of individuals), lgN is the decimal logarithm of N, log_2_N is the logarithm of N on base 2 (equivalent to 3.322 × lgN).

The results of the intervalization were adjusted to a uniform interval size of five days, and a series of phenotype frequencies/probabilities by lifespan was constructed and plotted on graphs. To construct a frequency series, we first found the number of individuals in a given interval (5 days). Then, we calculated the mortality rate in each interval as a percentage. Detailed instructions for dividing into intervals are provided in Appendix A, Steps 2–3. The intervalization process began on day zero, and the empty (zero) intervals at the beginning of the series were removed to eliminate their potential impact on the SW test results. The resulting frequency series were superimposed with either a normal distribution curve or β-distribution curve for illustrative purposes (see below).

To calculate the SW test, the samples were equalized by the number of intervals as follows:

First, the optimal interval size for each sample was calculated using Sturges’ rule (Formula (2)) to determine the number of intervals. If the number of intervals differed between the two samples, the smaller value was used in the calculations. The interval size for the second sample was recalculated by dividing its lifespan by the number of intervals of the first sample. Thus, while the interval size for each sample differs (intervals may be fractional), the number of intervals in both samples remains the same, allowing for correct comparisons using the SW test. If both genotypes had zero intervals at the beginning of the series (meaning there were no deaths at the start of the experiment), the zero intervals were removed to eliminate their potential impact on the SW test result. However, if a zero interval was present in only one sample, it was kept to maintain an equal number of intervals in both samples.

Second, the number of individuals falling into each interval for each sample was determined. This was done as follows: starting on day zero, the size of the interval found in the previous stage was measured. Next, the number of individuals whose lifespan was less than or equal to the obtained value was calculated. This process was repeated until the end of the lifespan series.

### 4.3. Formal Description of Frequency/Probability Series of Phenotypes by Lifespan and Mortality Series Using the Normal Distribution Function

To superimpose the normal distribution curve on the death per day series (Figure 1a), we calculated the value of the probability density function (PDF) for each day in the series (1, 2, 3, etc.) using the following formula:(3)$$F\left(x\right)=\frac{1}{\sigma \sqrt{2\pi }}e^{-\;\frac{{\left(x-µ\right)}^{2}}{{2\sigma }^{2}}}$$
where *µ* and *σ* are the mean and standard deviation, respectively, of the original (non-interval) lifespan series, *σ^2^* is the variance of lifespan, *e* is the base of the natural logarithm and π is the number Pi (3.14). The PDF can be calculated using the MS Excel function = NORM.DIST(x; µ; σ; FALSE), where FALSE is the PDF and TRUE is the cumulative distribution function (CDF).

*μ* and *σ* were calculated according to Formulas (4) and (5), respectively, with raw mortality data for lifespan series without interval partitioning (the value of σ calculated from interval data will be considerably wider than that calculated from non-interval data):(4)$$µ=\frac{\sum_{i=1}^{n}A_{i}*i}{N},$$(5)$$\sigma =\sqrt{\frac{\sum_{i=1}^{n}{{\left(i-µ\right)}^{2}*A}_{i}}{N-1}}$$
where $\sum_{i=1}^{n}$ is the sum from the first to the last day of measurement (*n*), A_i_ is the number of individuals that died on day *I*, *i* is the day on which the individual died (1, 2, 3, …) (actually, the individual’s lifespan), and *N* is the sample size (the total number of individuals).

The values of the normal distribution that were obtained (i.e., fractions of 1) were multiplied by the sample size (*N*). This was done when the normal distribution curve was superimposed on the graph showing mortality by day (Figure 1a). In this case, the scale of the normal distribution curve correlated well with the size of the bars (Figure 1a).

In cases where normal distribution curves were superimposed on the frequency series of phenotypes by lifespan, intervalized by five days, and graphs were constructed in MS Excel (Appendix A), the obtained values of the normal distribution function were multiplied by 100 (the sum of mortality frequencies across all intervals) and then by the interval size (five). In this case, *x* in Formula (3) was taken as the start day of the interval in the series of intervals (0, 5, 10…). This is related to a certain angularity of the curve in the graph in Appendix A. The start date of the interval was taken as x, since MS Excel does not allow two different X-axes to be combined on one graph (but it does allow two Y-axes to be combined). This means it is impossible to combine the frequency series (bars) and the normal distribution curve plotted by day (1, 2, 3, etc.).

For graphs constructed using a Python script (Figure 1c and subsequent figures of a similar type) the values of the normal distribution function obtained were multiplied by 100 (the sum of the mortality frequencies across all intervals) and then by the interval size (5). In this case, the scale of the normal distribution curve correlated well with the scale of the frequency series (Figure 1c and subsequent figures). For *x* in Formula (3), in this case, the sampling day (0, 1, 2, 3, etc.) was used.

The normal distribution function never reaches zero and approaches it infinitely; therefore, on graphs, the PDF of the normal distribution may appear as an arc-shaped curve “suspended above the X-axis in the air” without ever touching the X-axis.

### 4.4. Kolmogorov–Smirnov Test Calculation for Single Samples

The one-sample KS test was used to assess the normality of lifespan distributions. This test is based on identifying the supremum, or the point of maximum deviation between the experimental and expected cumulative distribution functions (CDFs), for a given sample [71,72] (Appendix A).

The CDF of the test was calculated using the following formula:(6)$$F\left(x\right)=\frac{1}{2}\left[1+\mathrm{erf}\left(\frac{x-µ}{\sigma \sqrt{2}}\right)\right]$$
where *F(x)* is the function of lifespan dependence on time, *x* is the lifespan of an individual, *µ* is the average life span, *σ* is the standard deviation of mean (the square root of the variance), and *erf* is the error function, which can be calculated using the formula:(7)$$\mathrm{erf}\left(z\right)=\frac{2}{\sqrt{\pi }}\int_{0}^{z}e^{{-t}^{2}}dt$$
where $\int_{0}^{z}$ is integral from 0 to *z*, π is the number π, dt is the infinitesimal increment of *t*, *z* is *z*-score (shows the deviation of the value *x* from *µ*, expressed as a quantity of σ), which can be calculated by the formula:(8)$$z=\frac{x-µ}{\sigma \sqrt{2}}$$

The differences between the experimental CDF (black stepped curve) and the expected CDF (gray straight line) (Appendix A) were found by subtracting the expected CDF values from the experimental CDF values at each point along the graph. The difference was considered positive when the experimental CDF deviated upward/leftward from the expected CDF and negative when it deviated downward/rightward. The maximum modulus of the difference between the experimental and expected CDFs was considered the supremum, or KS statistic, of the test (D_n_, D experimental). The calculated D_n_ values were then compared with the critical D-value (D*_crit_*) calculated for sample size *N* at the chosen significance level α (0.05). Critical values for *N* less than 35 were taken from Smirnov’s table [81]. For *N* between 35 and 100, critical values were taken from Miller’s table [82]. For *N* greater than 35, the formula d_α_/$\sqrt{n}$ was used, where d_α_(*N*) = 1.36 at α = 0.05. Values for other significance levels were taken from [67,83]. When D_n_ < D*_crit_*, the distribution was considered normal. If the exact sample size was unavailable in the tables, the *p*-value was found and compared to the chosen significance threshold, α. If the *p*-value was greater than α, the distribution was considered normal. We used the Kolmogorov’s formula to calculate the *p*-value, which is convenient for both small samples (intervalized lifespan series) (*N* < 35) and large samples (non-intervalized original lifespan series):(9)$$P=1-L\left(z\right),$$
where L(z):(10)$$L\left(z\right)=1-2*\sum_{y=1}^{\infty }(-{1)}^{y-1}*e^{-2*y^{2}*z^{2}}$$
where $\sum_{i=1}^{\infty }$ is the sum from 1 to ∞. In practice, the sum of the first ten values of *y* is significant; the rest do not add precision to the measurement; *e* is the base of the natural logarithm, $z=D_{n}*\sqrt{n},z^{2}={{(D}_{n})}^{2}*{\left(\sqrt{n}\right)}^{2}={{(D}_{n})}^{2}*n,$ where *n* is the sample size.

Survival curves were considered to be significantly different from each other at *p*-values less than 0.05.

The methodology for manually calculating the KS test on lifespan series for non-intervalized samples is provided in Appendix A. Explanations of the test can be found in classic papers [72,83]. To automate the calculations, we created a Python script. The script description is provided in Appendix A. The latest version of the script is available at: https://github.com/yumuhova/Distribution-analysis-of-the-lifespan-trait-in-Drosophila. URL accessed on 12 October 2025. We also attach a PDF file with the customized code for this script. When using the script to calculate the KS test for small samples and large samples, D_n_ is calculated using the classical methodology described in Appendix A and the *p*-value is calculated using the classical Kolmogorov’s formula.

### 4.5. Calculation of the Two Sample Kolmogorov–Smirnov Test

To assess the differences between the two survival curves, a two-sample KS test was used. The two-sample KS test calculation consists of three stages: (1) Find the lifespan as function of time (*S_t_*) for each sample, (2) Find the difference between the two *S_t_* functions (= D*_n,m_* statistic), (3a) Find D*_crit_* and compare it with the Smirnov’s percentage point table [81,82], or, instead, (3b) Find the *p*-value using Kolmogorov’s formula (= result of the test). Dividing the value obtained using Kolmogorov’s formula [71,72,73] by 2 gives the result of a one-sided test. However, there are other formulas for calculating the *p*-value of a one-sided test.

The values of the S_t_ function were found using the formula:(11)$$S\left(t\right)=\prod_{i=0}^{t}\frac{\left(n_{i}-d_{i}\right)}{n_{i}}$$
where $n_{i}$ is the number of individuals that lived until the day *t*, $d_{i}$ is the number of individuals that died at day *t*, $\prod_{i=0}^{t}$ is the result of multiplying all terms of the series from zero to the *t*-th term. In MS Excel, it is more convenient to find the value of the S*_t_* function for each day by multiplying the previously found value of the function (S*_t_*_−1_) by the value $\frac{\left(n_{t}-d_{t}\right)}{n_{t}}$ for day *t*.

The difference between the *S*_*t*1_ and *S*_*t*2_ functions was calculated using the formula:(12)$$D_{n,m}=\underset{t}{\sup}\left|S_{1,n}\left(t\right)-S_{2,m}\left(t\right)\right|$$
where *D_n_*_,*m*_ is the observed D (D statistics), *supS* is the maximum difference between subsets *S*_1_ and *S*_2_, $S_{1,n}\left(t\right)$ and $S_{2,m}\left(t\right)$ are the values of the *S_t_* function at each point of time *t* (day, in this case) for samples *n* and *m*, respectively.

The D*_crit_* was found using the following formula:(13)$$D_{crit}=d_{\alpha }(N)*\sqrt{\frac{n+m}{nm}}$$
where *n* and *m* этo are the volumes of the first and second samples, respectively, d_α_(*N*) is a tabulated value i.e., the coefficient used to find D*_crit_*.

The value of d_α_(*N*) is 1.358, 1.628, 1.731, 1.949 for significance levels α = 0.05, 0.01, 0.005, 0.001, respectively. The value of d_α_(*N*) can be found for any percentile point using the formula:(14)$$d_{\alpha }(N)=\sqrt{-\ln\left(\frac{\alpha }{2}\right)*\frac{1}{2}}$$

The null hypothesis (H0), which states that the survival curves do not differ, is rejected at a significance level of α when $D_{n,m}>D_{crit.}$ The exact *p*-value was calculated using Formulas (8) and (9). To find z, the following formula was used:(15)$$z=D_{n,m}*1/\sqrt{\frac{n+m}{nm}}$$

For a one-sided test, the *p*-value can be found using the following formula [84]:(16)$$P=e^{-2z^{2}-\frac{2z}{3}*\frac{m+2n}{\sqrt{mn\left(m+n\right)}}}$$
where *e* is the base of the natural logarithm, $z=D_{n}*\sqrt{n},\;z^{2}={{(D}_{n})}^{2}*{\left(\sqrt{n}\right)}^{2}={{(D}_{n})}^{2}*n$, where *n* and *m* are the sample sizes, *n* and *m*, respectively.

The attached MS Excel file on sheet Appendix A provides an example of instructions for calculating a two-sample KS test.

### 4.6. Shapiro-Wilk Test Calculation for Single Samples

The generalized SW test was used to evaluate the normality of lifespan distributions. Unlike the KS test, the SW test is based on a different principle and essentially evaluates the uniformity (graduality) of a data series (Appendix A). The coefficients used to calculate the SW test statistic (W) for small samples (≤50), i.e., intervalized lifespan series, were taken from Table 5 of the original paper [77]. Exact *p*-values for small samples were calculated using Royston’s method, which proposes calculating *p* forany sample size without a percentage point table. As an alternative, approximate *p*-values can be found using Table #6 from the original work by Shapiro and Wilk [77]. For large samples, i.e., non-intervalized lifespan series, both W and *p* were calculated using Royston’s method [69].

The *p*-value was calculated using the following final formula:(17)$$P=1-\Phi \left(z\right)$$
where Φ(z) is the standard (*µ* = 0, *σ* = 1) normal distribution function of *z*, *µ* is the mean lifespan, *σ* is the standard deviation (square root of the variance).

Φ(z) was found as follows:(18)$$\Phi \left(z\right)=\frac{1}{\sqrt{2\pi }}\int_{-\infty }^{z}e^{-\frac{t^{2}}{2}}dt$$
where *z* is the *z*-score or standard score, which shows the deviation of the value of *X* from *µ*, expressed as a quantity of *σ*, and *dt* is the infinitesimal increment of *t*.

*z* was found as follows:(19)$$z=\frac{\left(X-µ\right)}{\sigma }$$

The z-value can be found in the *z*-score table [85] or calculated using a normal distribution function. For example, the MS Excel function NORMDIST(X; µ; σ; TRUE) can be used, where TRUE is the cumulative function of the standard normal distribution (CDF).

The transformation of the observed z-distribution to the normal distribution (*µ* = 0, *σ* = 1) was performed using Royston’s formula:(20)$$z=\frac{\left(g\left(X\right)-µ-\sigma {µ}_{z}\right)}{\sigma \sigma_{z}}$$
where g(X) is the function that transforms the distribution of the original data into a normal distribution and µ_z_ and σ_z_ are the transformed µ and σ, respectively. Since µ = 0 and σ = 1, we obtain:(21)$$z=\frac{\left(g\left(X\right)-{µ}_{z}\right)}{\sigma_{z}}$$

The formulas and coefficients for calculating g(X), µ_z_, and σ_z_ according to Table 1 of Royston’s original paper [69] differ for samples of size less than 12 and for samples between 12 and 2000. We used the formulas and coefficients for samples ranging from 12 to 2000 (see below), given in Table 1 of Royston’s original paper, to calculate g(X), µ_z_ and σ_z_. We performed these calculations for all cases, i.e., when the sample size was >12 and when the sample size was <12 (11 in our case), since we found that using the formulas and coefficients for sample sizes ranging from 4 to 11 (as given in Royston’s original paper, Table 1) always yielded a *p*-value of zero. Therefore, the previously published formulas and coefficients are invalid for sample sizes ranging from 4 to 11.(22)$$g\left(X\right)=ln\left(1-W\right),$$(23)$${µ}_{z}\approx 0.0038915*\left({ln\left(n\right)}^{3}\right)-0.083751*\left(ln{\left(n\right)}^{2}\right)-0.31082*ln\left(n\right)-1.5861,$$(24)$$\sigma_{z}\approx e^{0.0030302*\left(ln{\left(n\right)}^{2}\right)-0.082676*\left(ln\left(n\right)\right)-0.4803}$$
where *n* is the sample size, *e* is the base of the natural logarithm, and *W* is the statistic of the SW test. “The numerator of W is proportional to the square of the ‘best’ (minimum variance, unbiased) linear estimator of the standard deviation, and the denominator is the sum of squares of the observations about the sample mean” [86].

The SW test statistic was calculated using the following formula from the original paper [77]:(25)$$W=\frac{{(\sum_{i=1}^{m}a_{n-i+1}*\;\left(x_{n-i+1}-x_{i}\right))}^{2}}{\sum_{i=1}^{n}({x_{i}-\frac{\sum x}{n})}^{2}}$$
where *x* is the sample, which is a sorted, ascending series of numbers showing the number of individuals that died in each interval, *n* is the sample size, or how many numbers are in the aforementioned series, *m* is n/2 for even *n* and (n − 1)/2 for odd n, and $\sum_{i=1}^{m}$ is the sum operator, where index *i* takes values from 1 to *m*. Xi is a member of the series, or sample, by count (1, 2, 3, etc.) and *a_n_*_−*i*+1_ are the coefficients from Table 5 for calculating the SW test statistic, as described in the original paper by Shapiro and Wilk [77]. W reflects the normality of the distribution. The value of W ranges from 0 to 1, and the closer W is to 1, the more probable it is that the data follow a normal distribution.

The methodology for the full manual calculation of the SW test on intervalized lifespan series is provided in Appendix A. Explanations of the SW test can be found in [69,77]. We created a Python script to automate the calculations and included a description in Appendix A. The program calculates *W* for small (intervalized) samples according to Shapiro and Wilk’s original paper and calculates the *p*-value according to Royston’s paper. For large (non-intervalized) samples, both *W* and *p*-value are calculated according to Royston’s paper.

### 4.7. Calculating the β-Distribution of Lifespan

The advantage of formally describing frequency series using the PDF of a β-distribution is that when calculating the PDF of a β-distribution, the first and last days of mortality are specified. This makes the graph of the PDF of a β-distribution coincide with the boundaries of the frequency series. Additionally, the first and last values of the β-distribution PDF are 0, unlike the normal distribution PDF, so the initial and final values of the β-distribution graph coincide with the X-axis. The normal distribution function approaches 0 infinitely and therefore extends beyond the boundaries of the graph, giving the appearance of being suspended in the air.

The PDF of the β-distribution was found using the formula from Section 18.4, (p. 244, [87]):(26)$$\beta \;distribution=\;\frac{x^{p-1}{\left(1-x\right)}^{q-1}}{\beta \left(p,q\right)}$$
where $\beta \left(p,q\right)$:(27)$$\beta \left(p,q\right)=\frac{\Gamma \left(p\right)\Gamma \left(q\right)}{\Gamma \left(p+q\right)}$$
where $\Gamma \left(z\right)$ is the gamma function, the condition of which is that *z* must be a rational positive number ($R\left(z\right)>0).$ The gamma function was found using the following formula:(28)$$\int_{0}^{\infty }t^{z-1}e^{-t}dt\;\;\;$$
where *t* are the values range that from zero to infinity on the X-scale, *z* is the argument for which the gamma function value is found, *e* is the base of the natural logarithm, and *dt* is the infinitesimal increment of *t*, the gamma function can be calculated using the =GAMMA(number) function in MS Excel, where (number) is the number on the X-axis for which the gamma function value is found on the Y-axis.

According to the law of β-distribution (Formula (26)) PDF is defined in the interval *x* < 1 and *x* > 0 (0 < *x* < 1). Two parameters, *p* and *q*, determine its shape, with *p* > 0, *q* > 0. The values of *p* and *q* can be expressed in consequently from the formulas for finding the Mean (*µ*) of the β-distribution and the Median (50th percentile of the sample, *Med*) of the β-distribution.

The following formula was used to calculate the *µ* of the β-distribution:(29)$$µ=\frac{p}{p+q}$$

In order to express *q* through *p* and *µ* a number of transformations were carried out, resulting in the following:(30)$$q=\;\frac{p}{µ}-p$$

In order to express *p* through *q* and *µ*, a number of transformations were carried out, resulting in the following:(31)$$p=\frac{µ*q}{1-µ}$$

The following formula was used to find the Median of β-distribution:(32)$$Med=\frac{p-\frac{1}{3}}{p+q-\frac{2}{3}}$$

Finally, to express *p* through the Median and *µ*, we substitute Formula (30) for *q* in Formula (32). This yields Formula (33), which enables us to find *p* given *µ* and *Med*:(33)$$p=\frac{1}{3}*\frac{1-2*Med}{1\;-\;\frac{Med}{µ}}$$

Next, we similarly express *q* through *Med* and *µ*:(34)$$q=\frac{1}{3}*\left(µ-1\right)*\frac{1-2*Med}{Med-µ}$$

As a result, knowing *Med* and *µ*, we can find *p* using the Formula (33) and *q* using the Formula (34). Then, we substitute the values of *p* and *q* into Formula (26), calculating it for values of x in the interval from 0 to 1.

In order to construct a β-distribution, the dimension of the distribution must first be normalized. Without normalization, the graph will not function properly since the β-distribution exists between 0 and 1. To normalize the dimension, subtract the first day of mortality from the µ values, then divide by the difference between the last and first days of mortality. For example, µ = 35 days and MaxLS = 84 days; the first death was recorded on day 8. Therefore, µ for constructing the PDF β-distribution = 35–8/(84–8).

In the simulations, the PDFs of the β-distribution were found using various specified values of Median and *µ*, or Mode and *µ*, ranging from 0 to 1. We used a decision tree to select between the Mode and the Median (Appendix A). Using only the Mode or Median to find *p* and *q* in the first approximation is more accurate from the perspective of experimental uniformity. However, it limits the range of possible distributions that can be represented, since some distributions can only be described using the Median. For example, the Mode cannot be used to describe a uniform distribution since the Mode value may be repeated several times in the lifespan series. Conversely, for some distributions, the Mode is more convenient since it is often located farther from µ, creating greater curvature of the graph and allowing it to better represent the frequency series.

If the equations *p* and/or *q* were calculated and the resulting values were less than 1 (the necessary condition for the equations to be valid is *p* ≥ 1 and *q* ≥ 1), then the standard values of *p* and *q*, which are both equal to 3.5, were used, resulting in a distribution that resembles a normal, bell-shaped curve.

In the Python script for visualizing the PDF of the β-distribution, we used µ from the original sample series. Depending on the decision tree, we used either the Mode or the Median from the original sample series or the intervalized series.

The exact decision tree algorithm for automatically constructing a β-distribution is as follows. First, the algorithm checks if a left- or right-sided β-distribution can be constructed using the Mode and, if not, the Median on the intervalized data. If *p* or q is less than 1, the algorithm proceeds to the initial variants of the series. If *p* or q is less than 1 in the initial variant, the algorithm adds 1 to the Initial_day indicator, the default value of which is 0 but can range from 0 to 4, until satisfactory values of *p* and q are found (*p* ≥ 1 and q ≥ 1). If no such values are found, then *p* and q are set to 3.5, and a bell-shaped distribution is plotted. Sometimes, however, the graphs produced by the automatic β-distribution construction Mode do not accurately represent the shape of the frequency series. In this case, the β-distribution shape selection section is applied (see the Python script).

The zero values of the β-distribution at its edges allow us to omit the edges of the distribution on the X-axis. The *µ* for the experimental sample was found using Formula (4). The following formula was used to find the Median (0.5):(35)$$F(m)=\int_{0}^{m}f\left(x\right)dx=0.5$$
where *m* is the Median (50th percentile of the sample), x is the upper limit of integration, f(x) is the function describing the lifespan at time *x*, and *dx* is an infinitesimal increment of *x*.

## 5. Conclusions

The present study offers a solution to the problem of representing survival data in a form alternative to conventional survival curves. A methodology for distribution assessment employing normality tests is proposed, complemented by visualization techniques based on the normal distribution function and the β-function. Representations of lifespan distributions were derived for laboratory lines of Drosophila melanogaster—including WT-like lines and WT-like lines subjected to a model genetic intervention (the white mutation). It has been demonstrated that genetic interventions can modify the frequencies of lifespan phenotypes, either by altering the distribution pattern or independently of such alterations.

The main results of the study can be summarized as follows:

1. Intervalized (e.g., 5-day) series of frequencies/probabilities of phenotypes by lifespan can be obtained from the initial lifespan series. Series of two samples superimposed on each other clearly show differences in the structure of the samples being compared. This method is an alternative to survival curves for displaying lifespan data. Dividing lifespan data into discrete intervals also minimizes the impact of daily mortality fluctuations (eliminating mortality gaps and outliers), making the data suitable for further analysis using the normality criterion.

2. The best conditions for analyzing lifespan data using the normality criterion are as follows: (1) use of the SW test on intervalized data for analysis (the SW test on original data, as well as the KS test on intervalized and original data, are not indicative) and (2) Intervalization of lifespan data according to Sturges’ rule with an equal number of intervals between the compared samples (the size of the interval between samples may differ) and a sample size of up to 200 individuals.

3. The frequency series of phenotypes by lifespan can be formally described using the PDF of a normal distribution (bell-shaped curve), or a PDF of a β-distribution (skewed, left- or right-sided curve, with a shifted center, or a symmetrical, bell-shaped curve, with parameters *p* and q equal to 3.5). This description allows for a quick and clear comparison of samples with each other (“at a glance”).

4. The distributions of the lifespan trait in laboratory (highly inbred) Drosophila lines are diverse, depend on the genotype of the line, and change when mutations are introduced. Often, the distributions of Drosophila lifespan do not meet the formal statistical criterion of normality of distribution. Bell-shaped distributions account for the majority of distributions. However, there are also distributions resembling uniform (plateau-like)/linear and left- and right-skewed β-distributions with a shifted center.

Thus, analyzing survival data using distribution analysis is more diverse and sensitive than the standard method of comparing samples based on median and maximum lifespan on survival curves. The proposed approach to analyzing lifespan distributions, and the use of the normality criterion for this purpose, could be valuable tools in lifespan and aging research, offering a new perspective on survival data. Understanding the distribution of phenotype frequencies by lifespan provides a more detailed picture of the processes affecting lifespan within the studied population, enabling more accurate tracking of the impact of genetic and pharmacological interventions on lifespan and ontogenesis.

## Figures and Tables

**Figure 1 ijms-26-11987-f001:**
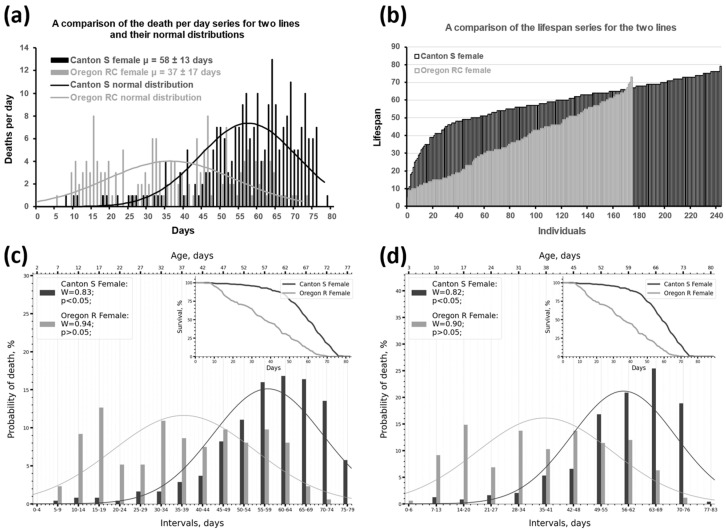
Original mortality distributions, mortality data converted to lifespan series and lifespan series converted to interval distributions/lifespan series. (**a**) Death-per-day data for females of two Drosophila WT-like laboratory lines with different genetic backgrounds (Canton S, black bars; Oregon RC, gray bars) are presented, along with normal distributions (bell-shaped curves) for these lines. Data from the w^67c23^ allele experiment (2012) were used for the analysis. The x-axis represents days (0, 1, 2, 3, etc.), and the y-axis represents the number of dead individuals per day. The numbers in the legend are µ (mean lifespan) ± standard deviation values. The normal distributions for the depicted death-per-day series were calculated as described in the Section 3.3 of the Materials and Methods. (**b**) The data from (**a**) were converted to a lifespan series. (**c**) The data from (**b**) were partitioned into five-day intervals using Sturges’ rule. The partitioning used was as follows: day 0–4, day 5–9, day 10–14, etc. Phenotype frequency distributions (bars) with superimposed normal distribution curves are shown. Each bar shows the probability of dying in a given five-day interval. Normal distributions were calculated as in (**a**). W is the SW test statistic and *p* is the SW test *p*-value. The inset in the upper right corner of the graph shows the survival curves for the sample data. This graph was obtained using the Python 3.11 script/program attached to the article (Appendix A). (**d**) The graph is similar to (**c**), but the data is divided into 7-day intervals. Data from (**a**) was used.

**Figure 2 ijms-26-11987-f002:**
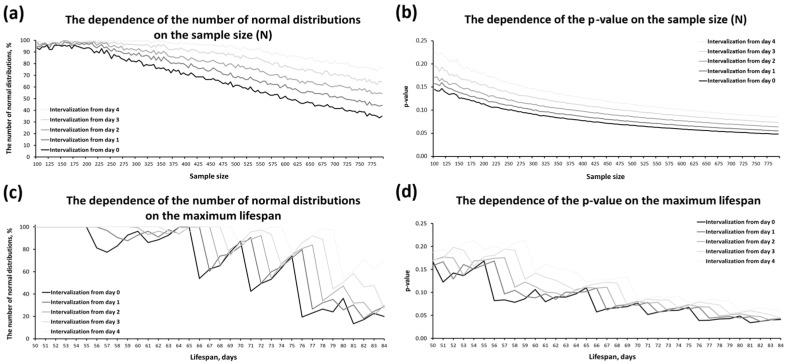
The figure shows the dependence of the frequency of preservation (retention) of normal distribution (**a**,**c**) and the *p*-value (**b**,**d**) on the sample size (number of individuals, *N*) (**a**,**b**) and the duration of lifespan (**c**,**d**) when performing the SW test. The dependencies were studied using an ideal normal series of lifespan of different lengths (different MaxLS) and with different *N*, intervalizing this series from the 0th, 1st, 2nd, 3rd, and 4th days (marked on the graphs with different shades of gray). The interval size was five days. For each point on the graph, the results were averaged across all samples with varying *N* (from 100 to 795 in increments of five individuals), lifespan duration (from 50 to 84 days, the range of lifespans recorded in our experiments, in increments of one day), and the first day of death (the first death was recorded on the first, second, third, or fourth day). A total of 19,600 simulated samples were used (140 *N* × 35 lifespans × 4 days of mortality onset).

**Figure 3 ijms-26-11987-f003:**
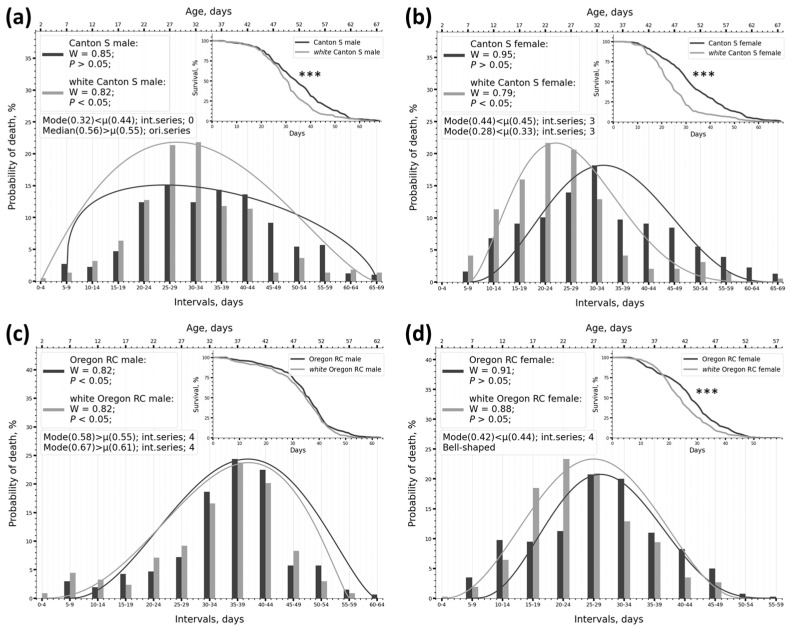
A comparison of the mutant and control in the experiment with allele w^1118^ (2024) is presented. (**a**,**b**) Canton S genetic background, (**c**,**d**) Oregon RC genetic background, (**a**,**c**) Comparison of WT males with *white* mutant males, (**b**,**d**) Comparisons of WT females with *white* mutant females. The bars show the phenotype frequencies/probabilities by lifespan, with β-distribution curves superimposed. w^1118^ and control flies of the same sex were compared. Frequency distributions were obtained by dividing the original lifespan series into 5-day intervals. The inset in the upper left corner shows the results of the SW test calculation. W is the Shapiro–Wilk (SW) test statistic and *p* is the SW test *p*-value. The inset in the upper right corner shows the survival curves for the compared samples. The genotypes are labeled in the legend. *** *p* < 0.001. The lower inset in the top left corner shows the parameters used to construct the β-distribution graph (black and grey arcs). int.series is the interval lifespan series, ori.series is the original lifespan series. The number (0–4) indicates the calendar day from which the β-distribution graph construction begins (see the Python script instruction in Appendix A and the Python code in the Appendix A).

**Figure 4 ijms-26-11987-f004:**
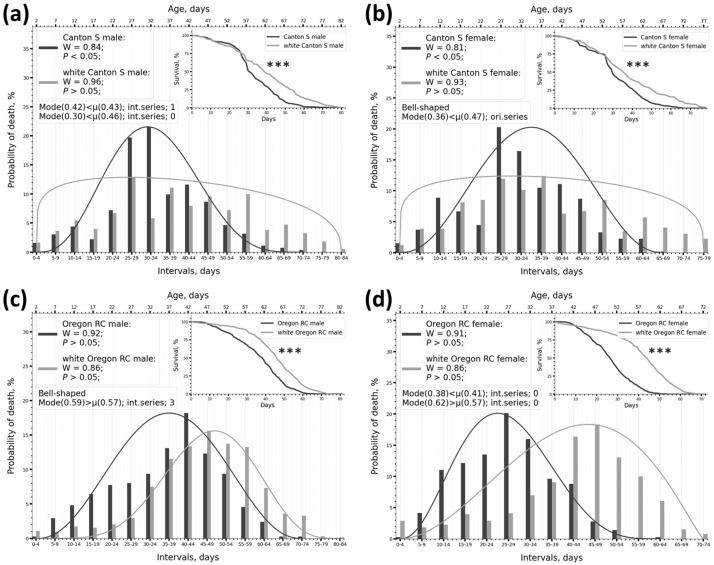
A comparison of the mutant and control in the experiment with allele w^67c23^ (2024) is presented. (**a**,**b**) Canton S genetic background, (**c**,**d**) Oregon RC genetic background, (**a**,**c**) Comparison of WT males with *white* mutant males, (**b**,**d**) Comparisons of WT females with *white* mutant females. The bars show the phenotype frequencies/probabilities by lifespan, with β-distribution curves superimposed. w^1118^ and control flies of the same sex were compared. Frequency distributions were obtained by dividing the original lifespan series into 5-day intervals. The inset in the upper left corner shows the results of the SW test calculation. W is the Shapiro–Wilk (SW) test statistic and *p* is the SW test *p*-value. The inset in the upper right corner shows the survival curves for the compared samples. The genotypes are labeled in the legend. *** *p* < 0.001. The lower inset in the top left corner shows the parameters used to construct the β-distribution graph (black and grey arcs). int.series is the interval lifespan series, ori.series is the original lifespan series. The number (0–4) indicates the calendar day from which the β-distribution graph construction begins (see the Python script instruction in Appendix A and the Python code in the Appendix A).

**Figure 5 ijms-26-11987-f005:**
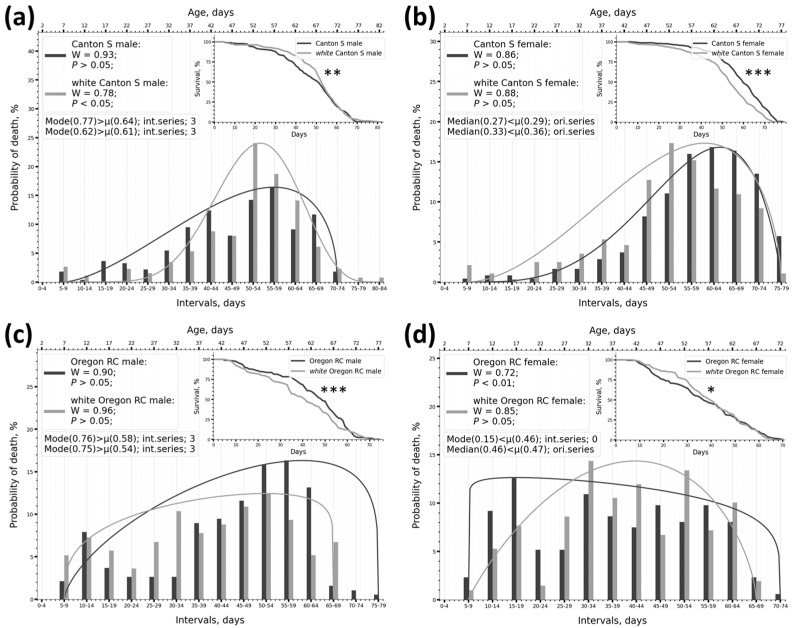
A comparison of the mutant and control in the experiment with allele w^67c23^ (2012) is presented. (**a**,**b**) Canton S genetic background, (**c**,**d**) Oregon RC genetic background, (**a**,**c**) Comparison of WT males with *white* mutant males, (**b**,**d**) Comparisons of WT females with *white* mutant females. The bars show the phenotype frequencies/probabilities by lifespan, with β-distribution curves superimposed. w^1118^ and control flies of the same sex were compared. Frequency distributions were obtained by dividing the original lifespan series into 5-day intervals. The inset in the upper left corner shows the results of the SW test calculation. W is the Shapiro–Wilk (SW) test statistic and *p* is the SW test *p*-value. The inset in the upper right corner shows the survival curves for the compared samples. The genotypes are labeled in the legend. *** *p* < 0.001. ** *p* < 0.01. * *p* < 0.05. The lower inset in the top left corner shows the parameters used to construct the β-distribution graph (black and grey arcs). int.series is the interval lifespan series, ori.series is the original lifespan series. The number (0–4) indicates the calendar day from which the β-distribution graph construction begins (see the Python script instruction in Appendix A and the Python code in the Appendix A).

**Table 1 ijms-26-11987-t001:** Lifespan measurement experiment with the w^1118^ allele (2024).

1	2	3	4	5	6	7	8	9	10	11	12	13	14	15
Genotype	Sex	µ, Days	σ, Days	SW Test w/o Intervals		SW Test Intervals		KS Test w/o Intervals		KS Test Intervals		Two Sample KS Test (WT vs. Mut.)	Number of Intervals	Sample Size
				W	*p*-Value *	W	*p*-Value	D_n_	*p*-Value	D_n_	*p*-Value			*N*
Canton S	M	34.8	± 12.7	0.99	0.036 not norm.	0.89	0.091 norm.	0.05	0.285 norm.	0.20	0.646 norm.	–	13	404
	F	33.1	± 13.5	0.98	0.001 not norm	0.94	0.478 norm.	0.07	0.091 norm.	0.15	0.917 norm.		13	308
w^1118^ Canton S	M	31.6	± 11.2	0.97	<0.0001 not norm.	0.80	0.005 not norm.	0.10	0.018 not norm.	0.26	0.307 not norm **	*p* < 0.001 ***	14	220
	F	25.2	± 10.7	0.92	<0.0001 not norm.	0.84	0.019 not norm.	0.12	0.011 not norm.	0.30	0.197 not norm **	*p* < 0.001	13	194
Oregon RC	M	35.2	± 10.4	0.96	<0.0001 not norm.	0.78	0.005 not norm.	0.10	<0.0001 not norm.	0.31	0.212 not norm **	–	12	472
	F	27.9	± 10.6	0.98	0.0001 not norm.	0.92	0.324 norm.	0.06	0.115 norm.	0.19	0.834 norm.		11	400
w^1118^ Oregon RC	M	33.7	± 10.9	0.95	<0.0001 not norm.	0.85	*0.034* *not norm.*	0.11	0.001 not norm.	0.20	*0.702* *norm*.	*p* > 0.05	12	337
	F	25.1	± 8.7	0.98	0.0004 not norm.	0.88	**0.106** **norm.**	0.08	**0.019** **not norm.**	0.21	0.725 norm.	*p* < 0.001	11	373

* at a significance threshold of *p*-value ≥ 0.05, the distribution was considered normal. ** not normal with Lilliefors adjustment for small sample sizes and normal without Lilliefors adjustment. *** the differences in survival curves between mutant and control flies of the same sex are shown.

**Table 2 ijms-26-11987-t002:** An experiment to measure lifespan with the w^67c23^ allele (2024).

1	2	3	4	5	6	7	8	9	10	11	12	13	14	15
Genotype	Sex	µ, Days	σ, Days	SW test w/o Intervals		SW Test Intervals		KS Test w/o Intervals		KS Test Intervals		Two Sample KS Test (WT vs. Mut.)	Number of Intervals	Sample Size
				W	*p*-Value *	W	*p*-Value	D_n_	*p*-Value	D_n_	*p*-Value			*N*
Canton S	M	33.3	± 12.7	0.98	<0.0001 not norm.	0.83	*0.010* *not norm.*	0.09	<0.0001 not norm.	0.22	*0.443* *norm.*	–	15	817
	F	31.2	± 13.3	0.99	<0.0001 not norm.	0.90	**0.123** **norm.**	0.11	**<0.0001** **not norm.**	0.18	0.740 norm.		14	676
w^67c23^ Canton S	M	39.7	± 18.2	0.99	0.0001 not norm.	0.97	** *0.756* ** ** *norm.* **	0.06	**0.020** **not norm.**	0.12	*0.960* *not norm.* **	*p* < 0.0001 ***	17	552
	F	37.5	± 18.0	0.98	<0.0001 not norm.	0.94	**0.408** **norm.**	0.07	**0.012** **not norm.**	0.18	0.654 norm.	*p* < 0.0001	16	493
Oregon RC	M	36.4	± 13.7	0.98	<0.0001 not norm.	0.94	**0.417** **norm.**	0.08	**0.013** **not norm.**	0.11	0.994 norm.	–	15	374
	F	26.7	± 10.5	0.99	0.0075 not norm.	0.92	0.232 norm.	0.05	0.254 norm.	0.17	0.839 norm.		13	363
w^67c23^ Oregon RC	M	45.6	± 14.2	0.97	<0.0001 not norm.	0.84	0.008 not norm.	0.06	0.019 not norm.	0.25	0.222 not norm **	*p* < 0.0001	17	641
	F	41.7	± 14.9	0.94	<0.0001 not norm.	0.87	0.039 not norm.	0.12	<0.0001 not norm.	0.22	0.479 not norm **	*p* < 0.0001	15	660

* at a significance threshold of *p*-value ≥ 0.05, the distribution was considered normal. ** not normal with Lilliefors adjustment for small sample sizes and normal without Lilliefors adjustment. *** the differences in survival curves between mutant and control flies of the same sex are shown.

**Table 3 ijms-26-11987-t003:** An experiment to measure lifespan with the w^67c23^ allele (2012).

1	2	3	4	5	6	7	8	9	10	11	12	13	14	15
Genotype	Sex	µ, Days	σ, Days	SW test w/o Intervals		SW Test Intervals		KS Test w/o Intervals		KS Test Intervals		Two Sample KS Test (WT vs. Mut.)	Number of Intervals	Sample Size
				W	*p*-Value *	W	*p*-Value *	D_n_	*p*-Value *	D_n_	*p*-Value *			*N*
Canton S	M	47.9	± 14.9	0.95	<0.0001 not norm.	0.92	**0.251** **norm.**	0.11	**0.003** **not norm**.	0.19	0.720 norm.	–	14	274
	F	58.3	± 13.2	0.92	<0.0001 not norm.	0.83	*0.010* *not norm.*	0.09	0.046 not norm.	0.22	*0.473* *norm.*		15	244
w^67c23^ Canton S	M	50.7	± 13.8	0.92	<0.0001 not norm.	0.80	0.003 not norm.	0.15	<0.0001 not norm.	0.22	0.406 not norm. **	*p* = 0.005 ***	16	262
	F	52.2	± 14.7	0.93	<0.0001 not norm.	0.89	**0.063** **norm.**	0.12	**0.001** **not norm**.	0.20	0.594 norm.	*p* < 0.0001	15	283
Oregon RC	M	44.6	± 16.5	0.92	<0.0001 not norm.	0.87	0.036 not norm.	0.13	0.003 not norm.	0.24	0.332 not norm. **	–	15	190
	F	37.2	± 17.4	0.96	<0.0001 not norm.	0.94	0.407 norm.	0.09	0.098 norm.	0.18	0.773 norm.		14	175
w^67c23^ Oregon RC	M	39.4	± 17.4	0.96	<0.0001 not norm.	0.98	**0.946** **norm.**	0.10	**0.043** **not norm.**	0.11	0.996 norm.	*p* < 0.001	13	193
	F	39.5	± 15.3	0.97	0.0002 not norm.	0.95	0.587 norm.	0.08	0.127 norm.	0.14	0.951 norm.	*p* = 0.040	14	209

* at a significance threshold of *p*-value ≥ 0.05, the distribution was considered normal. ** not normal with Lilliefors adjustment for small sample sizes and normal without Lilliefors adjustment. *** the differences in survival curves between mutant and control flies of the same sex are shown.

**Table 4 ijms-26-11987-t004:** Reproducibility of the method depending on the initial day of intervalization on experimental sample data.

1	2	3	4	5	6	7	8	9	11	12	13	14	15	16	17	18	19
Genotype	Sex	µ, Days	Internalization * from														
			Day 0			Day 1			Day 2			Day 3			Day 4		
			W	*p*- Value **	Number of Intervals	W	*p*-Value	Number of Intervals	W	*p*-Value	Number of Intervals	W	*p*-Value	Number of Intervals	W	*p*- Value	Number of Intervals
w^1118^ (2024)																	
Canton S	M	34.8	0.89	0.091	13	0.91	0.153	14	0.87	0.038	14	0.91	0.146	14	0.92	0.271	13
	F	33.1	0.94	0.478	13	0.93	0.262	14	0.94	0.457	14	0.92	0.242	14	0.93	0.282	14
w^1118^ Canton S	M	31.6	0.80	0.005	14	0.82	0.008	14	0.81	0.006	14	0.78	0.004	13	0.81	0.008	13
	F	25.2	0.84	0.019	13	0.83	0.017	13	0.81	0.009	13	0.80	0.006	13	0.79	0.005	13
Oregon RC	M	35.2	0.78	0.005	12	0.81	0.014	12	0.81	0.009	13	0.80	0.007	13	0.76	0.003	12
	F	27.9	0.92	0.324	11	0.93	0.400	11	0.95	0.654	11	0.92	0.294	12	0.92	0.289	12
*w^1118^* Oregon RC	M	33.7	0.85	0.034	12	0.85	0.034	12	0.83	0.023	12	0.81	0.017	11	0.82	0.020	11
	F	25.1	0.88	0.106	11	0.90	0.239	10	0.88	0.135	10	0.86	0.086	10	0.89	0.176	10
w^67c23^ (2024)																	
Canton S	M	33.3	0.83	0.010	15	0.77	0.002	15	0.80	0.004	15	0.87	0.029	15	0.87	0.039	14
	F	31.2	0.90	0.123	14	0.77	0.003	13	0.74	0.001	13	0.93	0.305	13	0.90	0.157	13
w^67c23^ Canton S	M	39.7	0.97	0.756	17	0.95	0.456	17	0.93	0.254	16	0.92	0.158	16	0.95	0.521	16
	F	37.5	0.94	0.408	16	0.96	0.699	16	0.98	0.965	16	0.95	0.433	16	0.96	0.713	15
Oregon RC	M	36.4	0.94	0.417	15	0.93	0.314	14	0.93	0.298	14	0.94	0.381	14	0.96	0.650	14
	F	26.7	0.92	0.232	13	0.91	0.222	12	0.93	0.393	12	0.90	0.161	12	0.92	0.267	12
*w^67c23^* Oregon RC	M	45.6	0.84	0.008	17	0.87	0.019	17	0.87	0.029	16	0.87	0.028	16	0.87	0.030	16
	F	41.7	0.87	0.039	15	0.87	0.035	15	0.88	0.055	15	0.86	0.027	14	0.89	0.088	14
w^67c23^ (2012)																	
Canton S	M	47.9	0.92	0.251	14	0.91	0.117	15	0.92	0.200	15	0.93	0.253	15	0.91	0.117	15
	F	58.3	0.83	0.010	15	0.82	0.007	15	0.80	0.004	15	0.83	0.009	15	0.81	0.004	16
*w^67c23^* Canton S	M	50.7	0.80	0.003	16	0.81	0.003	17	0.82	0.004	17	0.81	0.004	16	0.82	0.005	16
	F	52.2	0.89	0.063	15	0.89	0.067	15	0.84	0.010	16	0.84	0.008	16	0.87	0.032	15
Oregon RC	M	44.6	0.87	0.036	15	0.87	0.031	15	0.89	0.063	15	0.89	0.072	15	0.89	0.066	15
	F	37.2	0.94	0.457	14	0.94	0.438	14	0.97	0.906	14	0.95	0.637	14	0.91	0.143	14
*w^67c23^* Oregon RC	M	39.4	0.98	0.946	13	0.91	0.204	13	0.93	0.274	14	0.92	0.220	14	0.91	0.155	14
	F	39.5	0.95	0.587	13	0.92	0.290	13	0.95	0.608	13	0.96	0.772	12	0.92	0.287	13

* Interval length: 5 days. ** With a significance threshold of *p* ≥ 0.05, the distribution was considered normal. Green color in the table indicates normally distributed samples, red color indicates non-normally distributed samples.

**Table 5 ijms-26-11987-t005:** Comparison of the mutant and control when intervalizing according to Sturges’ rule (same number of intervals, different interval size) and intervalizing according to 5 days (different number of intervals, same interval size).

1	2	3	4	5	6	7	8	9	10	11
Genotype	Sex	µ, Days	Intervalization According to Sturges’s Rule				Intervalization Using a Five-Day Interval			
			W	*p*-Value *	Number of Intervals	Interval Length	W	*p*-Value	Number of Intervals	Interval Length
w^1118^ (2024)										
Canton S	M	34.8	0.85	0.079	9	7.17	0.89	0.091	13	5
	F	33.1	0.95	0.791	9	7.44	0.94	0.478	13	5
w^1118^ Canton S	M	31.6	0.82	0.038	9	7.29	0.80	0.005	14	5
	F	25.2	0.79	0.021	9	6.98	0.84	0.019	13	5
Oregon RC	M	35.2	0.82	0.030	10	6.07	0.78	0.005	12	5
	F	27.9	0.91	0.287	10	5.45	0.92	0.324	11	5
w^1118^ Oregon RC	M	33.7	0.82	0. 032	10	5.85	0.85	0.034	12	5
	F	25.1	0.88	0.148	10	5.03	0.88	0.106	11	5
w^67c23^ (2024)										
Canton S	M	33.3	0.84	0.032	11	7.12	0.83	0.010	15	5
	F	31.2	0.81	0.022	10	6.44	0.90	0.123	14	5
w^67c23^ Canton S	M	39.7	0.96	0.772	11	7.91	0.97	0.756	17	5
	F	37.5	0.93	0.445	10	7.74	0.94	0.408	16	5
Oregon RC	M	36.4	0.92	0.372	10	7.02	0.94	0.417	15	5
	F	26.7	0.91	0.261	10	5.89	0.92	0.232	13	5
w^67c23^ Oregon RC	M	45.6	0.86	0.087	10	8.38	0.84	0.008	17	5
	F	41.7	0.86	0.086	10	7.47	0.87	0.039	15	5
w^67c23^ (2012)										
Canton S	M	47.9	0.93	0.427	10	7.64	0.92	0.251	14	5
	F	58.3	0.86	0.109	9	7.95	0.83	0.010	15	5
w^67c23^ Canton S	M	50.7	0.78	0.011	10	8.52	0.80	0.003	16	5
	F	52.2	0.88	0.184	9	8.06	0.89	0.063	15	5
Oregon RC	M	44.6	0.90	0.235	9	8.17	0.87	0.036	15	5
	F	37.2	0.72	0.005	9	7.70	0.94	0.457	14	5
*w^67c23^* Oregon RC	M	39.4	0.96	0.878	9	7.35	0.98	0.946	13	5
	F	39.5	0.85	0.080	9	6.99	0.95	0.587	13	5

* With a significance threshold of *p* ≥ 0.05, the distribution was considered normal. Green color in the table indicates normally distributed samples, red color indicates non-normally distributed samples.

## Data Availability

The original contributions presented in this study are included in the article/Appendix A. Further inquiries can be directed to the corresponding authors.

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
