# Peer review of "Distribution Analysis of the Lifespan Trait in Drosophila"

_ijms, 2025, doi:10.3390/ijms262411987_

Round 1
Reviewer 1 Report
Comments and Suggestions for Authors
The manuscript "Distribution analysis of the lifespan trait in Drosophila" presents a novel methodological approach for the analysis of survival data. It proposes examining the distribution of lifespan as a quantitative trait, thereby moving beyond conventional comparisons of Kaplan-Meier curves. The authors provide a rigorous comparison of normality tests and establish clear, practical guidelines for applying their method. These include the use of the Shapiro-Wilk test on intervalised data and the critical importance of standardizing the number of intervals between compared samples. The demonstration that lifespan distributions in control lines are not necessarily normal and can be qualitatively altered by genetic interventions constitutes a significant finding. The provision of a Python script enhances the reproducibility of the study. The manuscript is recommended for publication after minor revisions addressing the points below.
Minor Comments
The Introduction currently focuses on the limitations of comparing survival curves using median/maximum lifespan and log-rank tests. To more fully establish the novelty and necessity of the proposed distribution analysis, the authors should briefly mention other established statistical approaches in the field. For instance, a short paragraph could reference methods such as Cox Proportional Hazards regression, Accelerated Failure Time (AFT) models, and analyses of mortality curve dynamics (e.g., using the Gompertz-Makeham law).
Briefly acknowledging these methods would allow the authors to more precisely define the specific gap their research aims to fill. This gap is the lack of focus on the intrinsic distribution of lifespan as a quantitative trait within a cohort, rather than merely the limitations of Kaplan-Meier curve comparisons. This will strengthen the study's motivation and demonstrate a comprehensive understanding of the methodological landscape.
Typos and Editorial Corrections
Lines 359 and 983: The text fragments "The text continues here." appear and should be removed, as they are likely artifacts from the Word template.
Figures 3-5: The labels for figure panels (a-d) are not described in the figure legends. A description for each subfigure should be provided.
Author Response
The authors heartily thank the reviewer for his (her) time and careful reading of the manuscript.
Here are our answers. Corrections in the text are marked in yellow.
Reviewer 1:
Minor Comments
The Introduction currently focuses on the limitations of comparing survival curves using median/maximum lifespan and log-rank tests. To more fully establish the novelty and necessity of the proposed distribution analysis, the authors should briefly mention other established statistical approaches in the field. For instance, a short paragraph could reference methods such as Cox Proportional Hazards regression, Accelerated Failure Time (AFT) models, and analyses of mortality curve dynamics (e.g., using the Gompertz-Makeham law).
Briefly acknowledging these methods would allow the authors to more precisely define the specific gap their research aims to fill. This gap is the lack of focus on the intrinsic distribution of lifespan as a quantitative trait within a cohort, rather than merely the limitations of Kaplan-Meier curve comparisons. This will strengthen the study's motivation and demonstrate a comprehensive understanding of the methodological landscape.
Response to the Minor Comments:
The authors added a paragraph about Cox's model, AFT model, and Gompertz's law in the introduction (lines 72-81).
Typos and Editorial Corrections
Comment 1 and 2:
Lines 359 and 983: The text fragments "The text continues here." appear and should be removed, as they are likely artifacts from the Word template.
Figures 3-5: The labels for figure panels (a-d) are not described in the figure legends. A description for each subfigure should be provided.
Response 1 and 2:
Both corrected.

Reviewer 2 Report
Comments and Suggestions for Authors
Dear authors,
Thanks for the opportunity to review the manuscript. This is a well-written and interesting manuscript. The proposed methodology for distribution assessment employing normality tests is of great significance for the research related to lifespan and aging. However, there are some writing, logical and format issues.
Specific Remarks:
Keywords: What is the difference between lifespan and longevity?
Line 54-57: These sentences have logical flaws. The author should reorganize them. However,... Thus? Rather?
Table 1, 2, 3: p should in italics. Please also check and revise the whole manuscript.
Line 768: add the space after ">", see also line 725,769, 837,753,754. To save everyone's time, I suggest authors carefully check and revise the whole manuscript.
Line 771: p should be in italics. See also Line 994. Please check carefully the whole manuscript.
Line 903: delete the period. See 954,982... Please read carefully the journal instructions for authors.
Line 915: you should provide a complete name when C. elegans was first mentioned.
Line 927-938: this paragraph is not suit for the Discussion part, I suggest moving it to the Introduction part.
Line 957: The figure citations are inconsistent in format. should be rewritten as: "Figs. 2, 3, and 4"
Line 971: “,” should revising "."
Line 975: delete “see”
Line 1001: To keep consistent, all letters for p should be uniformly in uppercase and lowercase. In addition, it should be in italics.
Line 1009: add the space between "N<100" see also line 1010.
Line 1074: Conclusions section is somewhat lengthy. It is difficult for readers to grasp your main conclusion. Please consider reorganizing and condensing this part.
Line 1094-1096: suggest using the abbreviation for Shapiro-Wilk test and Kolmogorov-Smirnov test, because this is not first mentioned.
Line 1096: 2)? Where is 1)?
In addition, there are many issues for reference lists:
Line 1168: This reference is incomplete, pages?
See also Line 1177, 1178,1197, 1200,
Line 1210: place published should be provided.
Line 1235: What is this?
Author Response
The authors heartily thank the reviewer for his (her) time and careful reading of the manuscript.
Here are our answers. Corrections in the text are marked in yellow.
Reviewer 2:
Specific Remarks:
What is the difference between lifespan and longevity?
Response to Specific Remarks:
Generally speaking, many scientists in the field perceive these two concepts as the same and use them interchangeably. However, in a narrower sense, they are different terms. In our understanding, lifespan is more about a trait that has a distribution, distribution ends, etc., and about how it changes. Longevity, on the other hand, is more about how long an organism can potentially live and how this can be achieved. This is how we understand it. This work is about lifespan.
Comment 1:
Line 54-57: These sentences have logical flaws. The author should reorganize them. However,... Thus? Rather?
Response 1:
The sentences have been rewritten and the logical inconsistencies were corrected (lines 82-86).
Comment 2:
Line 768: add the space after ">", see also line 725,769, 837,753,754. To save everyone's time, I suggest authors carefully check and revise the whole manuscript.
Response 2:
We carefully revised the whole manuscript and added spaces before and after each ">", "<" and "=" symbol.
Comment 3:
Table 1, 2, 3: p should in italics. Please also check and revise the whole manuscript.
Line 771: p should be in italics. See also Line 994. Please check carefully the whole manuscript.
Line 1001: To keep consistent, all letters for p should be uniformly in uppercase and lowercase. In addition, it should be in italics.
Response 3:
All instances of 'p' were italicized and capitalized, including the 'P' in 'p-value'.
Comment 4:
Line 903: delete the period. See 954,982... Please read carefully the journal instructions for authors.
Response 4:
Periods have been removed from the end of all subheadings.
Comment 5:
Line 915: you should provide a complete name when C. elegans was first mentioned.
Response 5:
Corrected.
Comment 6:
Line 927-938: this paragraph is not suit for the Discussion part, I suggest moving it to the Introduction part.
Response 6:
This paragraph was split into two parts. One part remained in the discussion, while the other was moved to the introduction section (lines 46-57).
Comment 7:
Line 957: The figure citations are inconsistent in format. should be rewritten as: "Figs. 2, 3, and 4"
Line 975: delete “see”
Response 7:
Citations of figures and tables have been edited, and “see” has been removed from them.
Comment 8:
Line 971: “,” should revising "."
Response 8:
Corrected.
Comment 9:
Line 1009: add the space between "N<100" see also line 1010.
Response 9:
We carefully revised the whole manuscript and added spaces before and after each ">", "<" and "=" symbol.
Comment 10:
Line 1094-1096: suggest using the abbreviation for Shapiro-Wilk test and Kolmogorov-Smirnov test, because this is not first mentioned.
Response 10:
We added the abbreviations for the Shapiro–Wilk and Kolmogorov–Smirnov tests. All subsequent instances were replaced with these abbreviations.
Comment 11:
Line 1096: 2)? Where is 1)?
Response 11:
“1)” was added.
Comment 12:
Line 1074: Conclusions section is somewhat lengthy. It is difficult for readers to grasp your main conclusion. Please consider reorganizing and condensing this part.
Response 12:
We agree that the сconclusions section is quite long. However, the authors kindly ask the reviewer to permit them to leave this section as it is, since the purpose of writing such a lengthy conclusion was to provide readers with a clear understanding of the paper's content without having to read the whole paper itself. As the paper is rather long and complex, some readers prefer to skip straight to the conclusion of such papers, to see what particular things were made.
Comment 13:
In addition, there are many issues for reference lists:
Line 1168: This reference is incomplete, pages?
See also Line 1177, 1178,1197, 1200,
Line 1210: place published should be provided.
Line 1235: What is this?
Response 13:
All corrected.

Round 2
Reviewer 2 Report
Comments and Suggestions for Authors
The authors have addressed all of my comments satisfactorily. Overall, the content of the manuscript shows a great deal of work, which is reflected in the quality of the current version. In my opinion, the manuscript can be accepted for publication after addressing some minor format issues. Congratulations to the authors!
Keywords: Author mentioned in the response letter: “This work is about lifespan”. In that case, is it necessary for "lifespan" and "longevity" to appear simultaneously?
References: Line 1352: "Mar 18" What is this? delete? See also Line 1192, 1253,1258,1280,1296,13311,1346, 1350, 1355.
Author Response
The authors thank the reviewer for his (her) work and discussion.
Here are our answers.
Reviewer 2:
Comments and Suggestions for Authors
The authors have addressed all of my comments satisfactorily. Overall, the content of the manuscript shows a great deal of work, which is reflected in the quality of the current version. In my opinion, the manuscript can be accepted for publication after addressing some minor format issues. Congratulations to the authors!
Keywords: Author mentioned in the response letter: “This work is about lifespan”. In that case, is it necessary for "lifespan" and "longevity" to appear simultaneously?
Response to Comments and Suggestions for Authors
The authors thank the reviewer for his (her) attention to detail. Yes, we would like to keep both in order to increase the number of people who pay attention to the article. We think it is important to attract researchers working in the field of longevity, studying individual genes or mutations, etc., so that they turn their attention to lifespan distributions. Understanding exactly how and in response to what lifespan distributions change may be the key to increasing human longevity. At least, that is what we currently think.
Comment 1:
References: Line 1352: "Mar 18" What is this? delete? See also Line 1192, 1253,1258,1280,1296,13311,1346, 1350, 1355.
Response 1:
All corrected.
